# Test-Time Graph Neural Dataset Search With Generative Projection

Xin Zheng [1]   Wei Huang [2]   Chuan Zhou [3]   Ming Li [4]   Shirui Pan [1]

## Abstract

In this work, we address the test-time adaptation challenge in graph neural networks (GNNs), focusing on overcoming the limitations in *flexibility* and *generalization* inherent in existing methods. To this end, we propose a novel research problem, **test-time graph neural dataset search**, which learns a parameterized test-time graph distribution to enhance the inference performance of unseen test graphs on well-trained GNNs. Specifically, we propose a generative **P**rojection based test-time **G**raph **N**eural **D**ataset **S**earch method, named **PGNDS**, which maps the unseen test graph distribution back to the known training distribution through a generation process guided by well-trained GNNs. The proposed PGNDS framework consists of three key modules: (1) *dual conditional diffusion* for GNN-guided generative projection through test-back-to-training distribution mapping; (2) *dynamic search* from the generative sampling space to select the most expressive test graphs; (3) *ensemble inference* to aggregate information from original and adapted test graphs. Extensive experiments on real-world graphs demonstrate the superior ability of our proposed PGNDS for test-time GNN inference.

## 1. Introduction

Graph neural networks (GNNs) have demonstrated impressive success in learning from diverse graph-structured data across a wide range of real-world applications, such as recommender systems, traffic forecasting, and drug discovery (Koh et al., 2024; Zhang et al., 2024a; Zheng et al., 2022a;b; 2023c;a; Jin et al., 2022; Zheng et al., 2022c; 2023d; Luo et al., 2023; Zheng et al., 2024b; 2025; Yu et al., 2025; Zhang et al., 2024b). Despite the delicate design and thorough training of GNNs on training graphs (Pan et al., 2023; Wang et al., 2024), these models often face significant performance degradation during test-time inference on test graphs in the deployment stage (Wu et al., 2022; Liu et al., 2023; Chen et al., 2023b; Yu et al., 2023; Huang et al., 2025; Zhang et al., 2025). The main cause of this degradation is the *distribution shifts* between the training and test graphs, which often occur due to substantial changes in node contexts, edge connections, and graph-level distributions at test time (Wu et al., 2022; Liu et al., 2023; Chen et al., 2023b; Yu et al., 2023; Zhang et al., 2025).

To tackle the issue of test-time performance degradation caused by distribution shifts, test-time adaptation (TTA) has recently gained attention as a promising strategy for enhancing inference performance on test graphs (Chen et al., 2022; Wang et al., 2022; Jin et al., 2023; Zhang et al., 2024e;d). Typically, TTA on graphs aims to dynamically fine-tune well-trained GNN models or modify test graphs to improve GNN model generalization and inference performance. Based on the adaptation objective—whether adapting the model or the data—existing methods can be classified into two categories: (a) test-time model adaptation (Chen et al., 2022; Wang et al., 2022; Zhang et al., 2024e); and (b) test-time graph adaptation (Jin et al., 2023). Specifically, given the unseen test graphs, test-time model adaptation mainly works on updating the well-trained GNN models using the self-supervised learning paradigm, where the primary objective is to optimize or fine-tune the pretrained GNN model parameters. In contrast, test-time graph adaptation takes a data-centric approach, modifying the test graph data while keeping the well-trained GNN model parameters unchanged.

However, these two types of methods, as shown in Fig. 1 (a) and (b), face significant challenges in the real world: **Challenge 1: Impracticality of fine-tuning deployed GNNs.** Once deployed, fine-tuning GNNs for adaptation is impractical due to online application constraints and the high computational cost of updating large models. **Challenge 2: Inefficiency of data-centric graph adaptation.** While test-time graph adaptation avoids modifying model parameters,

---

[1]School of Information and Communication Technology, Griffith University, Gold Coast, Australia. [2]RIKEN Center for Advanced Intelligence Project, Tokyo, Japan. [3]Academy of Mathematics and Systems Science, Chinese Academy of Sciences, Beijing, China. [4]Zhejiang Institute of Optoelectronics, Jinhua, China; Zhejiang Key Laboratory of Intelligent Education Technology and Application, Zhejiang Normal University, Jinhua, China. Correspondence to: Shirui Pan <s.pan@griffith.edu.au>.

*Proceedings of the $42^{nd}$ International Conference on Machine Learning*, Vancouver, Canada. PMLR 267, 2025. Copyright 2025 by the author(s).

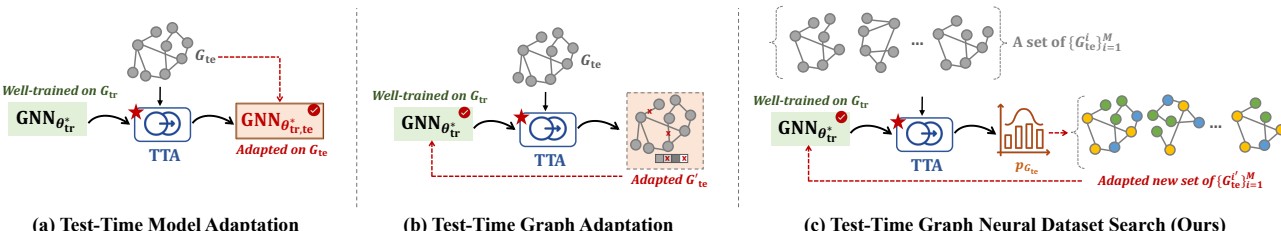

**(a) Test-Time Model Adaptation**    **(b) Test-Time Graph Adaptation**    **(c) Test-Time Graph Neural Dataset Search (Ours)**

*Figure 1.* Comparison among test-time adaptation (TTA) techniques on graphs: (a) test-time model adaptation: requires model fine-tuning and uses the adapted GNN for inference; (b) test-time graph adaptation: requires individually-parameterized graph transformation and uses the original well-trained GNN for inference; and (c) our proposed test-time graph neural dataset search: captures parameterized test graph dataset distribution and uses the original well-trained GNN for inference. The red dashed line indicates the final inference process.

existing methods like GTRANS (Jin et al., 2023) are limited in *flexibility* and *generalization*, requiring node- and edge-level adjustments and per-graph fine-tuning, making them inefficient for diverse graph distributions.

In light of these, we propose a fresh perspective for test-time adaptation on graphs, where the key learning objective for the data-centric solution can be the answer of following:

> **Question:** Rather than modifying individually-parameterized test graphs, can we learn a parameterized test-time graph distribution, to project the entire test graph set distribution back to the training set?

The underlying intuition is that, in the ideal scenario without distribution shifts, a well-trained model would perform optimally on unseen test graphs. This implies that if the test graph can be effectively mapped back to the training graph in a data-centric transformation process, the generalization error can be minimized to the greatest extent.

To answer this question, in this work, we propose a new research problem, **test-time graph neural dataset search**, which aims to learn the optimal distribution parameters for unknown test graph datasets, enabling them to generalize effectively on well-trained GNN models during test-time inference. Specifically, as shown in Fig. 1 (c), test-time graph neural dataset search transforms unknown and distribution-shifted test graphs into a new test graph distribution, tailored to specific GNN model architectures, learning behaviors, and task objectives of well-trained GNNs. According to the learned optimal distribution, this approach automatically generates a new test graph dataset, without modifying the well-trained GNN model parameters but enabling better adaptation performance.

To solve this new research problem, we propose a generative **P**rojection based test-time **G**raph **N**eural **D**ataset **S**earch method, named **PGNDS**, to learn the optimal test graph distribution driven by a score-based diffusion model, by capturing test-back-to-training distribution with a GNN-guided generative projection process. Specifically, given a set of test graphs with unknown labels and potential distribution

shifts, the proposed PGNDS framework consists of three essential phases: (1) *dual conditional diffusion*, which incorporates well-trained GNN model-specific information, including task-specific graph representations and labels, as dual conditions to guide the score updates in the reverse process of the score-based diffusion model. This phase ensures that the refined test graphs preserve their essential data characteristics while becoming more aligned with the training distribution, leading to more effective test-time inference performance. (2) *dynamic search*, which spans the entire reverse process and leverages the learned test graph distribution to identify the most expressive test graphs, leading to a renewed and adapted test graph set. (3) *ensemble inference*, which aggregates the representative information from both the original and adapted test graphs, enabling the well-trained GNN to more accurate prediction results on the unknown and distribution-shifted test graph set. As a result, the proposed PGNDS could achieve flexible and well-generalized test-time adaptation with the automatic generation of renewed test graphs, enhancing the test-time inference performance in a data-centric manner. In summary, the contributions of this work are listed as follows:

- **New Research Problem.** We introduce a new research problem, *test-time graph neural dataset search*, to learn the optimal distribution of unknown test graph datasets for effective adaptation on well-trained GNN models during test-time inference.

- **Graph Data-Centric Solution.** We develop a generative projection based test-time graph neural dataset search method, named PGNDS, to learn the optimal test graph distribution driven by a diffusion model. By projecting test graphs back to the training distribution, PGNDS enables effective test-time adaptation by automatically generating refined test graphs.

- **Comprehensive Experiments.** We evaluate the proposed PGNDS on real-world graph datasets, and extensive experimental results demonstrate its superior test-time adaptation ability for inference on well-trained GNN models.

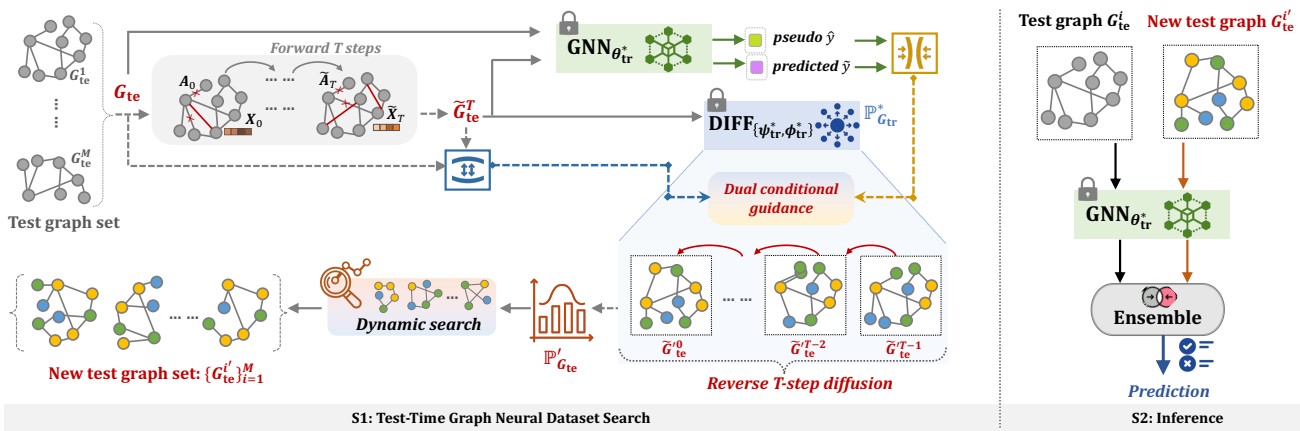

*Figure 2.* Overall framework of our proposed generative projection based test-time graph neural dataset search method (PGNDS).

**Related Work.** Our research aligns with prior work on *test-time adaptation (TTA)* on graph (Jin et al., 2023; Zheng et al., 2023b; Chen et al., 2023a; Jang et al., 2022), which focuses on dynamically adjusting pre-trained models or data to enhance adaptation to test samples. More detailed related work of *TTA* and *other distribution shift related problem* can be found in Appendix A.

## 2. The Proposed Method

### 2.1. Problem Definition

**Notations.** Given a training graph set $\mathcal{G}_{\text{tr}} = \{G_{\text{tr}}^1, G_{\text{tr}}^2, \cdots, G_{\text{tr}}^N\} \sim \mathbb{P}_{\mathcal{G}_{\text{tr}}}$ containing $N$ graphs, where each graph $G_{\text{tr}}^i = (\mathbf{X}_{\text{tr}}^i, \mathbf{A}_{\text{tr}}^i, y_{\text{tr}}^i)$, where $\mathbf{X}_{\text{tr}}^i$ denotes the node feature matrix, $\mathbf{A}_{\text{tr}}^i$ is the adjacency matrix indicating whether nodes are connected or not, $y_{\text{tr}}^i$ denotes the graph label, and $\mathbb{P}_{\mathcal{G}_{\text{tr}}}$ is the training graph distribution. Similarly, the test graph set can be represented as $\mathcal{G}_{\text{te}} = \{G_{\text{te}}^1, G_{\text{te}}^2, \cdots, G_{\text{te}}^M\} \sim \mathbb{P}_{\mathcal{G}_{\text{te}}}$ containing $M$ graphs, where each graph $G_{\text{te}}^i = (\mathbf{X}_{\text{te}}^i, \mathbf{A}_{\text{te}}^i)$ without labels and $\mathbb{P}_{\mathcal{G}_{\text{te}}}$ is the test graph distribution. Typically, during test-time inference, there exist unknown and distribution shifts between the training and test graph distributions, *i.e.*, $\mathbb{P}_{\mathcal{G}_{\text{te}}} \neq \mathbb{P}_{\mathcal{G}_{\text{tr}}}$.

**Preliminary.** This work focuses on addressing the test-time adaptation problem on graphs. During test time, a GNN model has been pre-trained on a training graph set using a standard supervised learning approach. As part of our proposed method, which incorporates a generative projection technique, we outline the training procedures for both the GNN model and the graph generative model (*i.e.*, diffusion model) as essential preliminary steps.

■ **GNN Model Training.** A GNN model is trained on the training graph set $\mathcal{G}_{\text{tr}}$ for graph-level classification or regres-

sion tasks as follows:

$$\boldsymbol{\theta}_{\text{tr}}^* = \min_{\boldsymbol{\theta}_{\text{tr}}} \mathcal{L}_{\text{GNN}}\left(\hat{\mathbf{Y}}_{\text{tr}}, \mathbf{Y}_{\text{tr}}\right), \text{ where}$$
$$\mathbf{Z}_{\text{tr}}, \hat{\mathbf{Y}}_{\text{tr}} = \text{GNN}_{\boldsymbol{\theta}_{\text{tr}}}(\mathcal{G}_{\text{tr}}). \tag{1}$$

Here, $\boldsymbol{\theta}_{\text{tr}}$ denotes GNN parameters, $\mathbf{Z}_{\text{tr}} \in \mathbb{R}^{N \times d_1}$ is the output graph embeddings, and $\hat{\mathbf{Y}}_{\text{tr}} = \{\hat{y}_{\text{tr}}^1, \hat{y}_{\text{tr}}^2, \cdots, \hat{y}_{\text{tr}}^N\} \in \mathbb{R}^{N \times 1}$ denotes the output graph labels predicted by the trained $\text{GNN}_{\boldsymbol{\theta}_{\text{tr}}}$. According to different graph learning tasks, $\mathcal{L}_{\text{GNN}}$ can be the cross-entropy loss for graph classification, or the mean squared error loss for graph regression, where $\mathbf{Y}_{\text{tr}}$ is the ground-truth graph labels. The well-trained GNN model can be denotes as $\text{GNN}_{\boldsymbol{\theta}_{\text{tr}}^*}$ with optimal weight parameters $\boldsymbol{\theta}_{\text{tr}}^*$.

■ **Graph Diffusion Model Training.** We focus on the score-based generative graph diffusion model (Jo et al., 2022) to capture the distribution of the training graph set. Given the training graph set $\mathcal{G}_{\text{tr}}$, the diffusion process can be represented as the trajectory of random variables $\{G_{\text{tr}}^t = (\mathbf{X}_{\text{tr}}^t, \mathbf{A}_{\text{tr}}^t)\}_{t \in [0, T_{\text{tr}}]}$, where $G_{\text{tr}}^0 \in \mathcal{G}_{\text{tr}} \sim \mathbb{P}_{\mathcal{G}_{\text{tr}}}$ and $G_{\text{tr}}^{T_{\text{tr}}}$ follows a prior Gaussian distribution. Then, the diffusion process solves the following stochastic differential equation (SDE) from the graph to the noise:

$$\mathrm{d}G_{\text{tr}}^t = \mathbf{f}\left(G_{\text{tr}}^t, t\right) \mathrm{d}t + g\left(t\right) \mathrm{d}\mathbf{w}, \quad G_{\text{tr}}^0 \sim \mathbb{P}_{\mathcal{G}_{\text{tr}}}, \tag{2}$$

where $\mathbf{f}(\cdot, t) : \mathcal{G} \to \mathcal{G}$ denotes the linear drift coefficient, $g(t) : \mathbb{R} \to \mathbb{R}$ denotes the diffusion coefficient, and $\mathbf{w}$ is the standard Wiener process. After such a forward process, which adds noise to the original graph distribution to transform it into a Gaussian distribution, the reverse-time SDE for diffusion can be presented as:

$$\mathrm{d}G_{\text{tr}}^t = \left[\mathbf{f}\left(G_{\text{tr}}^t, t\right) - g_t^2 \nabla_{G_{\text{tr}}^t} \log p_t\left(G_{\text{tr}}^t\right)\right] \mathrm{d}\bar{t} + g_t \, \mathrm{d}\bar{\mathbf{w}}, \tag{3}$$

where $p_t$ represents the marginal distribution at the time $t$ under the forward diffusion process, $\bar{\mathbf{w}}$ is a reverse-time standard Wiener process, and $\mathrm{d}\bar{t}$ is an infinitesimal negative time step. To solving the reverse diffusion

process $\nabla_{G_{\mathrm{tr}}^t} \log p_t(G_{\mathrm{tr}}^t)$, the time-dependent score-based method (Jo et al., 2022) trains $\mathbf{s}_{\psi_{\mathrm{tr}}}(\cdot, t)$ for node features and $\mathbf{s}_{\phi_{\mathrm{tr}}}(\cdot, t)$ for structures until optimal as following:

$$
\begin{aligned}
\mathbf{s}_{\psi_{\mathrm{tr}}^*}\left(G_{\mathrm{tr}}^t, t\right) &\approx \nabla_{\mathbf{X}_t} \log p_t\left(G_{\mathrm{tr}}^t\right), \\
\mathbf{s}_{\phi_{\mathrm{tr}}^*}\left(G_{\mathrm{tr}}^t, t\right) &\approx \nabla_{\mathbf{A}_t} \log p_t\left(G_{\mathrm{tr}}^t\right).
\end{aligned} \tag{4}
$$

Hence, $\psi^*$ and $\phi^*$ are well-trained diffusion model parameters with GNN backbones. It is important to note that all pre-trained parameters from both GNN and diffusion models will remain *frozen* throughout the entire procedure of our proposed method. Specifically, the goal of the proposed test-time graph neural dataset search can be described as:

**Definition 2.1** (**Test-time Graph Neural Dataset Search**). Given the test graph set $\mathcal{G}_{\mathrm{te}} = \{G_{\mathrm{te}}^1, G_{\mathrm{te}}^2, \cdots, G_{\mathrm{te}}^M\}$ and the well-trained $\mathrm{GNN}_{\boldsymbol{\theta}_{\mathrm{tr}}^*}$ model, test-time graph neural dataset search aims to learn the new adapted test graph set $\mathcal{G}_{\mathrm{te}}'$, searched from a projected distribution $\mathbb{P}_{\mathcal{G}_{\mathrm{te}}}'$ with a parameterized mapping function $\mathcal{F}_{\boldsymbol{\Omega}}(\cdot)$ as:

$$
\mathcal{G}_{\mathrm{te}}' \sim J(\mathbb{P}_{\mathcal{G}_{\mathrm{te}}}'|\epsilon), \text{ where } \mathbb{P}_{\mathcal{G}_{\mathrm{te}}}' = \mathcal{F}_{\boldsymbol{\Omega}}(\mathcal{G}_{\mathrm{te}}, \mathrm{GNN}_{\boldsymbol{\theta}_{\mathrm{tr}}^*}). \tag{5}
$$

In Eq. (5), $J(\cdot|\epsilon)$ denotes the search function with $\epsilon$ as the certain conditional criterion, and $\boldsymbol{\Omega}$ represents the parameters of the mapping function that requires be optimized.

### 2.2. Overview of the Proposed PGNDS

The overall framework of our proposed PGNDS for test-time graph neural dataset search is presented in Fig. 2. Generally, the framework consists of two stages: S1: test-time graph neural dataset search, which mainly covers two sub-steps: (S1-1): dual conditional diffusion and (S1-2): dynamic search; S2: test-time ensemble inference. Specifically, for (S1-1) dual conditional diffusion, given a set of test graphs $\mathcal{G}_{\mathrm{te}} = \{G_{\mathrm{te}}^1, G_{\mathrm{te}}^2, \cdots, G_{\mathrm{te}}^M\}$, we take an arbitrary $G_{\mathrm{te}}$ omitted superscript as an illustration example. It first experiences $T$-step forward process by adding Gaussian perturbation to both node features and edge connections. Then, the perturbed graph $\widetilde{G}_{\mathrm{te}}^T$ is simultaneously fed into the well-trained GNN model $\mathrm{GNN}_{\boldsymbol{\theta}_{\mathrm{tr}}^*}$ for pseudo label $\widetilde{y}$ and the well-trained diffusion model $\mathrm{DIFF}_{\{\psi_{\mathrm{tr}}^*, \phi_{\mathrm{tr}}^*\}}$ for reverse diffusion process, where the reverse diffusion process is guided by dual conditional information from $\mathrm{GNN}_{\boldsymbol{\theta}_{\mathrm{tr}}^*}$. For (S1-2) dynamic search, given the learned test graph distribution $\mathbb{P}_{\mathcal{G}_{\mathrm{te}}}'$, we can obtain a set of potential new test graphs, guided by task-relevant information from $\mathrm{GNN}_{\boldsymbol{\theta}_{\mathrm{tr}}^*}$. Then, a dynamic search module is employed to identify the most expressive test graph by spanning the entire diffusion process, leading to a renewed and adapted test graph set. For (S2) ensemble inference, with the adapted test graph set, we introduce an ensemble scheme to aggregate the representative information from both the original and adapted test graphs. More details of the proposed framework are presented as follows.

### 2.3. Modular Design

**Dual Conditional Diffusion.** For an arbitrary $G_{\mathrm{te}}^i \in \mathcal{G}_{\mathrm{te}}$, its state at time step $t$ is denoted as $G_{\mathrm{te}}^t$, with the subscript $i$ omitted for simplicity. The dual conditional diffusion phase first conducts the forward perturbation, which slightly perturbs the original test graph set to diversify the graph characteristics of both nodes and structures for better back-to-training projection. Hence, the test-time forward process can be denoted as:

$$
\mathrm{d}G_{\mathrm{te}}^t = \mathbf{f}\left(G_{\mathrm{te}}^t, t\right) \mathrm{d}t + \mathbf{g}(t)\,\mathrm{d}\mathbf{w}, \quad G_{\mathrm{te}}^0 \sim \mathbb{P}_{\mathcal{G}_{\mathrm{te}}}, \tag{6}
$$

where $\{G_{\mathrm{te}}^t = (\mathbf{X}_{\mathrm{te}}^t, \mathbf{A}_{\mathrm{te}}^t)\}_{t \in [0, T]}$, following the same process as Eq. (2) but with $T \ll T_{\mathrm{tr}}$. This constraint helps prevent over-projection of noise, ensuring that the perturbed test graphs remain within the learning space of the well-trained diffusion model.

After this, we obtain a set of perturbed test graphs $\{\widetilde{G}_{\mathrm{te}}^t = (\widetilde{\mathbf{X}}_{\mathrm{te}}^t, \widetilde{\mathbf{A}}_{\mathrm{te}}^t)\}_{t \in [T, 0]}$, along with the original clear test graph $G_{\mathrm{te}}^0$, for which are fed into the well-trained $\mathrm{GNN}_{\boldsymbol{\theta}_{\mathrm{tr}}^*}$ as:

$$
\widetilde{y}_{\mathrm{te}}^t = \mathrm{GNN}_{\boldsymbol{\theta}_{\mathrm{tr}}^*}(\widetilde{\mathbf{X}}_{\mathrm{te}}^t, \widetilde{\mathbf{A}}_{\mathrm{te}}^t), \quad \hat{y}_{\mathrm{te}}^0 = \mathrm{GNN}_{\boldsymbol{\theta}_{\mathrm{tr}}^*}(\mathbf{X}_{\mathrm{te}}^0, \mathbf{A}_{\mathrm{te}}^0). \tag{7}
$$

This step provides the predicted label $\widetilde{y}_{\mathrm{te}}^t$ in the reverse-time diffusion and the *pseudo* ground-truth label $\hat{y}_{\mathrm{te}}^0$ to capture the GNN model-specific and graph learning task-driven information from the well-trained GNN.

Then, taking this (a) task-specific information and (b) graph data-centric constraint between the $(\widetilde{G}_{\mathrm{te}}^t, G_{\mathrm{te}}^0)$ as dual conditional guidance, the reverse-time diffusion process with the well-trained $\mathrm{DIFF}_{\{\psi_{\mathrm{tr}}^*, \phi_{\mathrm{tr}}^*\}}$ can be written as

$$
\begin{cases}
\mathrm{d}\widetilde{\mathbf{X}}_{\mathrm{te}}^t = \left[\mathbf{f}_1(\widetilde{\mathbf{X}}_{\mathrm{te}}^t, t) - g_{1,t}^2\left(\mathbf{s}_{\psi_{\mathrm{tr}}^*}(\widetilde{G}_{\mathrm{te}}^t, t) - \mathbf{s}_{\mathrm{te}}^{\mathrm{feat}}\right)\right] \mathrm{d}\bar{t} + g_{1,t}\mathrm{d}\overline{\mathbf{w}}_1, \\
\mathrm{d}\widetilde{\mathbf{A}}_{\mathrm{te}}^t = \left[\mathbf{f}_2(\widetilde{\mathbf{A}}_{\mathrm{te}}^t, t) - g_{2,t}^2\left(\mathbf{s}_{\phi_{\mathrm{tr}}^*}(\widetilde{G}_{\mathrm{te}}^t, t) - \mathbf{s}_{\mathrm{te}}^{\mathrm{struc}}\right)\right] \mathrm{d}\bar{t} + g_{2,t}\mathrm{d}\overline{\mathbf{w}}_2,
\end{cases} \tag{8}
$$

where $\mathbf{s}_{\mathrm{te}}^{\mathrm{feat}}$ and $\mathbf{s}_{\mathrm{te}}^{\mathrm{struc}}$ denote the test-time score rectification functions in terms of node features and graph structure, respectively, and they can be calculated as follows:

$$
\begin{aligned}
\mathbf{s}_{\mathrm{te}}^{\mathrm{feat}}(\widetilde{y}_{\mathrm{te}}^t, \hat{y}_{\mathrm{te}}, \widetilde{G}_{\mathrm{te}}^t, G_{\mathrm{te}}^0) &= \alpha \cdot \mathbf{r}_{\mathrm{gtask}}(\widetilde{y}_{\mathrm{te}}^t, \hat{y}_{\mathrm{te}}) + \gamma \cdot \mathbf{r}_{\mathrm{gdist}}(\widetilde{G}_{\mathrm{te}}^t, G_{\mathrm{te}}^0), \\
\mathbf{s}_{\mathrm{te}}^{\mathrm{struc}}(\widetilde{\mathbf{A}}_{\mathrm{te}}^t, \mathbf{A}_{\mathrm{te}}^0) &= \beta \cdot \mathbf{r}_{\mathrm{struc}}(\widetilde{\mathbf{A}}_{\mathrm{te}}^t, \mathbf{A}_{\mathrm{te}}^0).
\end{aligned} \tag{9}
$$

where $\alpha$, $\beta$, and $\gamma$ are the hyper-parameters to control the relative importance of three distinct conditional constraints: $\mathbf{r}_{\mathrm{gtask}}(\cdot)$, $\mathbf{r}_{\mathrm{struc}}(\cdot)$, and $\mathbf{r}_{\mathrm{gdist}}(\cdot)$, which guide the generation process from the view of: graph learning tasks, graph data-centric diversity, and graph structures, respectively, for rectifying the reverse scores on test graphs. More specifically, these three rectification functions can be defined as follows:

$$
\begin{aligned}
\mathbf{r}_{\mathrm{gtask}}\left(\hat{y}_{\mathrm{te}} \mid \widetilde{y}_{\mathrm{te}}^t\right) &= -\log p\left(\hat{y}_{\mathrm{te}} \mid \widetilde{y}_{\mathrm{te}}^t\right), \tag{10} \\
\mathbf{r}_{\mathrm{struc}}\left(\widetilde{\mathbf{A}}_{\mathrm{te}}^t, \mathbf{A}_{\mathrm{te}}^0\right) &= \mathcal{D}_{\mathrm{MSE}}\left(\widetilde{\mathbf{A}}_{\mathrm{te}}^t, \mathbf{A}_{\mathrm{te}}^0\right), \\
\mathbf{r}_{\mathrm{gdist}}\left(\widetilde{G}_{\mathrm{te}}^t, G_{\mathrm{te}}^0\right) &= \mathcal{D}_{\mathrm{FGW}}\left(\widetilde{G}_{\mathrm{te}}^t, G_{\mathrm{te}}^0\right),
\end{aligned}
$$

where $\mathcal{D}_{\mathrm{MSE}}(\cdot, \cdot)$ denotes the mean squared error, and $\mathcal{D}_{\mathrm{FGW}}(\cdot, \cdot)$ denotes the fused Gromov-Wasserstein distance for defining a distance metric on the input graph space during test-time on the well-trained GNN model, which excels at determining the optimal coupling between nodes in a fused metric space.

Specifically, for task specific rectification function $\mathbf{r}_{\mathrm{gtask}}$, it uses *pseudo* ground-truth label $\hat{y}_{\mathrm{te}}$ as the condition to guide the diffusion model in generating a test graph that corresponds to $\hat{y}_{\mathrm{te}}$, capturing the task-driven test graph distribution, ensuring the generated new test graph can sufficiently preserve the label of the original graph during the reverse-time diffusion. For graph structure rectification function $\mathbf{r}_{\mathrm{struct}}$, it calculates the graph structural distance to enforce that the generated graph structure does not deviate significantly from the original one, so that critical connections between nodes are preserved. For graph diversity rectification function $\mathbf{r}_{\mathrm{gdist}}$, it leverages global information by capturing the interaction between the graph structure and node features, through maximizing the distance between the generated test graph and the original graph, encouraging diversity to prevent the generated graph from collapsing into the original test graph.

**Dynamic Search.** After completing the $T$-step reverse time of the proposed dual conditional diffusion, the proposed PGNDS captures a new condition-guided test-time graph distribution $\mathbb{P}'_{\mathcal{G}_{\mathrm{te}}}$ for adapting the well-trained GNN models. Unlike the standard reverse diffusion, which retains only the final denoised graph, in this work, we build a graph dataset search space that spans all reverse steps within the new test graph distribution, so we have:

$$\widetilde{\mathcal{G}}'_{\mathrm{te}} = \{\widetilde{G}'^{(i,t)}_{\mathrm{te}} | i = 1, \cdots, M; t \in [T, 0]\} \sim \mathbb{P}'_{\mathcal{G}_{\mathrm{te}}}. \quad (11)$$

Now the key is to search with a function $J(\cdot|\epsilon)$ according to Eq. (5) for the best-adapted test graphs from the learned distribution $\mathbb{P}'_{\mathcal{G}_{\mathrm{te}}}$. This process is guided by a specific criterion $\epsilon$ for graph data selection, which is designed by incorporating three rectification constraints as outlined below:

$$\epsilon^{i,*} = \min_{t} \left[ \alpha \cdot \mathbf{r}^{(i,t)}_{\mathrm{gtask}} + \beta \cdot \mathbf{r}^{(i,t)}_{\mathrm{struc}} - \gamma \cdot \mathbf{r}^{(i,t)}_{\mathrm{gdist}} \right]_{t \in [T,0]}. \quad (12)$$

That means, for each test graph $\widetilde{G}'^{(i,t)}_{\mathrm{te}}$, we select the optimal adapted graph with the smallest test-time rectification effort on three-fold constraints at a certain $t$-step as $G'^{i}_{\mathrm{te}}$. Additionally, to enhance the efficiency of the search process, we introduce a dynamic stopping condition for the reverse diffusion. Specifically, suppose the overall distance does not decrease after a certain number of patience steps. In that case, the reverse diffusion process is halted, and the best-adapted graph with the current minimum $\epsilon^{i,*}$.

**Ensemble Inference.** After the dynamic search from the learned test-time graph distribution with $J(\mathbb{P}'_{\mathcal{G}_{\mathrm{te}}}|\epsilon)$,

we could obtain a new adapted test graph set as $\mathcal{G}'_{\mathrm{te}} = \{G'^1_{\mathrm{te}}, G'^2_{\mathrm{te}}, \cdots, G'^M_{\mathrm{te}}\}$. This adapted set is aligned with the training distribution while retaining the maximum characteristics of the test distribution. However, given the inherently complex nature of the test-time graph distribution, it is crucial to ensure that the generation process can maximally approximate its ground truth. Hence, even with the integration of multiple conditional guidance mechanisms, the search space for identifying the *optimal* set of adapted test graphs remains extensive. In light of this, we propose an ensemble scheme to aggregate the information from the original test graph $G_{\mathrm{te}}$ and the adapted test graph $G'_{\mathrm{te}}$. We adopt the representation ensemble scheme for the graph regression task and the prediction ensemble scheme for the graph classification task.

Specifically, given $\mathbf{z}_{\mathrm{te}}, \hat{y}_{\mathrm{te}} = \mathrm{GNN}_{\boldsymbol{\theta}^*_{\mathrm{tr}}}(G_{\mathrm{te}})$ and $\mathbf{z}'_{\mathrm{te}}, \hat{y}'_{\mathrm{te}} = \mathrm{GNN}_{\boldsymbol{\theta}^*_{\mathrm{tr}}}(G'_{\mathrm{te}})$, $\mathbf{z}_{\mathrm{te}}$ and $\mathbf{z}'_{\mathrm{te}}$ are graph embeddings. For the graph classification task, the final prediction $\mathbf{y}_{\mathrm{cls}}$ of the well-trained GNN during the test time is the average confidence of both graphs. For the graph regression task, the final output $\mathbf{y}_{\mathrm{reg}}$ is generated using the fused graph embeddings before being input into the predictor of the well-trained GNN. Hence, we have:

$$\begin{aligned} \mathbf{y}_{\mathrm{cls}} &= \arg\max_{c} \frac{1}{2} \left( \hat{y}^c_{\mathrm{te}} + \hat{y}'^c_{\mathrm{te}} \right), \\ \mathbf{y}_{\mathrm{reg}} &= f_{\boldsymbol{\theta}^*_{\mathrm{tr}}} \left( \eta \cdot \mathbf{z}_{\mathrm{te}} + (1 - \eta) \, \mathbf{z}'_{\mathrm{te}} \right), \end{aligned} \quad (13)$$

where $c \in \{1, \cdots, C\}$ and $C$ denotes the number of label classes, and $f_{\boldsymbol{\theta}^*_{\mathrm{tr}}}(\cdot)$ is the predictor of the well-trained $\mathrm{GNN}_{\boldsymbol{\theta}^*_{\mathrm{tr}}}$. The proposed ensemble scheme in our PGNDS enable the automatic selection of how much to weigh the original and adapted test graphs, leading to better robustness of the test-time inference on the well-trained GNN model.

## 2.4. Theoretical Justification

We provide a theoretical analysis demonstrating the feasibility of our proposed dual conditional diffusion in guiding the effective generation of new test graph sets. This approach ensures that the generated test graphs fall into a mixed distribution $\mathbb{P}'_{\mathcal{G}_{\mathrm{te}}}$ that combines the training $\mathbb{P}_{\mathcal{G}_{\mathrm{tr}}}$ and test graph set distributions $\mathbb{P}_{\mathcal{G}_{\mathrm{te}}}$, $\mathbb{P}_{\boldsymbol{\theta}^*_{\mathrm{tr}}}$ from $\mathrm{GNN}_{\boldsymbol{\theta}^*_{\mathrm{tr}}}$, and $\mathbb{P}_{\{\boldsymbol{\psi}^*_{\mathrm{tr}}, \boldsymbol{\phi}^*_{\mathrm{tr}}\}}$ from $\mathrm{DIFF}_{\{\boldsymbol{\psi}^*_{\mathrm{tr}}, \boldsymbol{\phi}^*_{\mathrm{tr}}\}}$.

**Proposition 2.2.** *Given the training graph distribution $\mathbb{P}_{\mathcal{G}_{tr}}$ and the test graph distribution $\mathbb{P}_{\mathcal{G}_{te}}$, the complete forward-reverse diffusion process begins with $G^{t=0}_{te}$ for $T \ll T_{tr}$ steps and ends with a potential pseudo back-to-training test graph distribution $\widetilde{G}'^{0}_{te} \sim \hat{\mathbb{P}}'_{\mathcal{G}_{te}}$. This process corresponds to a multiple integral of multi-step transition probabilities, lacking a closed-form solution and without guaranteeing distribution alignment.*

*Proof.* We primarily use the node feature $\mathbf{X}^0_{\mathrm{te}} \sim \mathbb{P}_{\mathcal{G}_{\mathrm{te}}}$ for

*Table 1.* ROC-AUC performance (↑) comparison between baseline methods and our proposed PGNDS on molecular and protein graphs for the graph classification task. Best results are in bold, and the second-bests are underlined.

| Methods | Molecular | | | Protein |
|---|---|---|---|---|
| | Ogbg-BBBP | Ogbg-BACE | Ogbg-ClinTox | Enzymes |
| ERM (Wu et al., 2022) | 0.6935 | 0.8265 | 0.6271 | 0.7161 |
| EERM (Wu et al., 2022) | 0.6857 | 0.7738 | 0.6314 | 0.5274 |
| TENT (Wang et al., 2020) | 0.6988 | 0.8338 | 0.6254 | 0.7187 |
| DropEdge (Rong et al., 2019) | 0.6950 | 0.8139 | 0.6256 | 0.7137 |
| FeatMask (You et al., 2020) | 0.6973 | 0.8240 | 0.6298 | 0.7115 |
| GTRANS (Jin et al., 2023) | 0.6921 | 0.8256 | 0.6312 | 0.5037 |
| **PGNDS (Ours)** | **0.7014** | **0.8873** | **0.6371** | **0.7245** |

*Table 2.* RMSE performance (↓) comparison between baseline methods and our proposed PGNDS on molecular graphs for the graph regression task. Best results are in bold, and the second-bests are underlined. GTRANS method with '-' on QM9-alpha indicates the resulted RMSE exceeding 1000+, signifying unstable optimization.

| Methods | QM9 | | | | Ogbg-FreeSolv |
|---|---|---|---|---|---|
| | A | B | C | alpha | |
| ERM (Wu et al., 2022) | 2.2022 | 0.4415 | **0.3924** | 4.3220 | 1.9308 |
| DropEdge (Rong et al., 2019) | 2.5879 | 0.5439 | 0.3978 | 9.8058 | 2.4434 |
| FeatMask (You et al., 2020) | 2.3560 | 0.5183 | 0.3991 | 8.3528 | 2.6432 |
| GTRANS (Jin et al., 2023) | 2.8192 | 5.3664 | 0.5183 | - | 1.9289 |
| **PGNDS (Ours)** | **2.1985** | **0.4283** | 0.3926 | **4.2881** | **1.9252** |

proof illustration (simplified as $\mathbf{x}_0 \sim q(\mathbf{x}_0)$ in the following), with the structure adhering to the same principles. Then, the forward process can be described as:

$$\mathbf{x}_0 \sim q(\mathbf{x}_0),\ \mathbf{x}_1 \sim q(\mathbf{x}_1|\mathbf{x}_0),\ \ldots,\ \mathbf{x}_T \sim q(\mathbf{x}_T|\mathbf{x}_{T-1}).$$
(14)

Thus, at step $T$, the distribution of $\mathbf{x}_T$ is:

$$q_T(\mathbf{x}_T) = \int q(\mathbf{x}_0) \prod_{t=1}^{T} q(\mathbf{x}_t|\mathbf{x}_{t-1})\, d\mathbf{x}_0 \ldots d\mathbf{x}_{T-1}. \quad (15)$$

Next, instead of using the reverse process matching $q$, we employ a reverse transformation derived from the well-trained diffusion model $\text{DIFF}_{\psi_{\text{tr}}^*}$ with score-based approximation, denoting as $p_{\psi}(\mathbf{x}_{t-1}|\mathbf{x}_t)|_{t=T}^{1}$. This reverse sampling takes $\mathbf{x}_T$ step-by-step back to $\mathbf{x}_0$. Thus, the complete forward-reverse combination can be viewed as a unified Markov chain within combined $2T$ steps:

$$\mathbf{x}_0 \xrightarrow{q} \mathbf{x}_1 \xrightarrow{q} \ldots \xrightarrow{q} \mathbf{x}_T \xrightarrow{p_{\psi}} \mathbf{x}_{T-1} \xrightarrow{p_{\psi}} \ldots \xrightarrow{p_{\psi}} \mathbf{x}_0'. \quad (16)$$

Here, the final output $\mathbf{x}_0'$ may not necessarily equal the original $\mathbf{x}_0$. Then, its distribution can be expressed as:

$$\hat{\mathbb{P}}_{\mathcal{G}_{\text{te}}}'(\mathbf{x}_0') = \int \cdots \int q(\mathbf{x}_0) \left[ \prod_{t=1}^{T} q(\mathbf{x}_t \mid \mathbf{x}_{t-1}) \right]$$
$$\times \left[ \prod_{t=T}^{1} p_{\psi}(\mathbf{x}_{t-1}' \mid \mathbf{x}_t') \right] \delta(\mathbf{x}_0' - \mathbf{x}_0)\, d\mathbf{x}_0\, d\mathbf{x}_1 \cdots d\mathbf{x}_T\, d\mathbf{x}_{T-1}' \cdots d\mathbf{x}_1'. \quad (17)$$

For simplicity, we treat $\mathbf{x}_t' \simeq \mathbf{x}_t$ for effectively moving the notation $\mathbf{x}_T'$ back to $\mathbf{x}_T$ and projecting to a single $\mathbf{x}_0$. Theoretically, this is a nested Markov chain with multiple integrals of multi-step transition probabilities. The obtained pseudo test distribution $\hat{\mathbb{P}}_{\mathcal{G}_{\text{te}}}'$ differs from the original $q$ (*i.e.*, $\mathbb{P}_{\mathcal{G}_{\text{te}}}$) or $p_{\psi}$ (*i.e.*, $\mathbb{P}_{\mathcal{G}_{\text{tr}}}$), it is a *hybrid* product, where $\mathbb{P}_{\mathcal{G}_{\text{te}}}$ is used in the forward direction and $\mathbb{P}_{\mathcal{G}_{\text{tr}}}$ in the reverse direction. If $\mathbb{P}_{\mathcal{G}_{\text{te}}} \neq \mathbb{P}_{\mathcal{G}_{\text{tr}}}$ and $\text{DIFF}_{\psi_{\text{tr}}^*}$ is not perfectly optimized, we can conclude that the final distribution $\hat{\mathbb{P}}_{\mathcal{G}_{\text{te}}}' \neq \{\mathbb{P}_{\mathcal{G}_{\text{te}}}, \mathbb{P}_{\mathcal{G}_{\text{tr}}}\}$.
□

**Proposition 2.3.** *Given the absence of a closed-form solution and the high degree of freedom of the hybrid pseudo test distribution $\hat{\mathbb{P}}_{\mathcal{G}_{te}}'$ without any guidance, the proposed dual conditional guidance in* PGNDS *could effectively mitigate distribution misalignment through $\mathbb{P}_{\mathcal{G}_{te}}'$, by constraining the deviation within a controllable bound $\xi$, encapsulated by the well-trained GNN parameter distribution $\mathbb{P}_{\boldsymbol{\theta}_{tr}^*}$.*

*Proof.* After introducing guidance terms, the reverse diffusion $p_{\psi}(x_{t-1}'|x_t')$ in Eq. (17) can be modified as:

$$p_{\{\psi, \boldsymbol{\theta}_{\text{tr}}^*\}}(\mathbf{x}_{t-1}'|\mathbf{x}_t') = p_{\psi}(\mathbf{x}_{t-1}'|\mathbf{x}_t')$$
$$\times \exp\left( \alpha \cdot \mathbf{r}_{\text{gtask}}(\mathbf{x}_t') + \gamma \cdot \mathbf{r}_{\text{struc}}(\mathbf{x}_t') + \beta \cdot \mathbf{r}_{\text{gdist.}}(\mathbf{x}_t') \right),$$
(18)

Then, during each step of the reverse process, guidance terms apply gradient constraints to the generated distribu-

tion:

$$\nabla \log p_{\{\boldsymbol{\psi}, \boldsymbol{\theta}_{\text{tr}}^*\}}(\mathbf{x}_{t-1}'|\mathbf{x}_t') = \nabla \log p_{\boldsymbol{\psi}}(\mathbf{x}_{t-1}'|\mathbf{x}_t') \\ + \sum_{i \in \{\alpha, \beta, \gamma\}} i \nabla \mathbf{r}_i \cdot (\mathbf{x}_t'). \quad (19)$$

Hence, given $\boldsymbol{\omega}(\cdot)$ represents the distribution discrepancy measurement function (*e.g.*, Wasserstein distance). By incorporating the guidance terms, the bias of the integral formula can be expressed as a reduction in distribution bias as:

$$\boldsymbol{\omega}(\mathbb{P}_{\mathcal{G}_{\text{te}}}', p_{\{\boldsymbol{\psi}, \boldsymbol{\theta}_{\text{tr}}^*\}}) \leq \boldsymbol{\omega}(\hat{\mathbb{P}}_{\mathcal{G}_{\text{te}}}', p_{\boldsymbol{\psi}}) - \eta \cdot |\xi|, \quad \text{where} \\ |\xi| = \sum_{t=1}^{T} \left\| \nabla \left( \alpha \cdot \mathbf{r}_{\text{gtask}}^{(t)} + \gamma \mathbf{r}_{\text{struc}}^{(t)} + \beta \mathbf{r}_{\text{gdist}}^{(t)} \right) \right\|^2, \quad (20)$$

where $\eta > 0$ is the learning rate and $|\xi|$ is the controllable bound under our proposed guidance terms. $\square$

## 3. Experiments

We verify the effectiveness of the proposed PGNDS in terms of the test-time inference performance. Concretely, we aim to answer the following questions: **Q1:** How does the proposed PGNDS perform on the well-trained GNN for both graph classification and regression tasks when faced with unknown graph distribution shifts at test time? **Q2:** How does the proposed PGNDS perform in ablation studies focusing on each components? **Q3:** How sensitive is the proposed PGNDS to variations in hyper-parameters? **Q4:** How does the proposed PGNDS perform in terms of running time efficiency?

### 3.1. Experimental Settings

**Datasets & Metrics.** We perform experiments on six real-world graph datasets covering protein and molecular graphs, with four graph classification tasks and five graph regression tasks. We use the area under the ROC curve (ROC-AUC) to evaluate the graph classification task and root mean square error (RMSE) for the graph regression task. Higher ROC-AUC ($\uparrow$) and lower RMSE ($\downarrow$) indicate better graph learning performance. More details of datasets are listed in Appendix B. Note that the original QM9 dataset contains nineteen tasks, but we selected four of them (*i.e.*, A, B, C, and alpha) for our experiments. For all training and test graphs, we follow the process procedures and splits in previous works (Liu et al., 2024; Jo et al., 2022).

**Baseline Methods & Implementation.** We compare the proposed PGNDS with the following baselines that fall in two groups: *graph model-centric methods*: empirical risk minimization (ERM) for standard training (Wu et al., 2022), data augmentation technique DropEdge (Rong et al., 2019) and FeatMask (You et al., 2020), Explore-to-Extrapolate Risk Minimization (EERM) (Wu et al., 2022) customized

*Table 3.* Modular indexes for ablation study in our PGNDS.

| Modules | Dual Conditional Diffusion | | | | Dynamic Search | Ensemble Inference |
|---|---|---|---|---|---|---|
| | Guidance | $\mathbf{r}_{\text{gtask}}$ | $\mathbf{r}_{\text{struc}}$ | $\mathbf{r}_{\text{dist}}$ | | |
| Idx01 | ✗ | ✗ | ✗ | ✗ | ✗ | ✗ |
| Idx02 | ✓ | ✓ | ✗ | ✗ | ✗ | ✗ |
| Idx03 | ✓ | ✓ | ✓ | ✗ | ✗ | ✗ |
| Idx04 | ✓ | ✓ | ✓ | ✓ | ✗ | ✗ |
| Idx05 | ✓ | ✓ | ✓ | ✓ | ✓ | ✗ |
| Idx06 (Ours) | ✓ | ✓ | ✓ | ✓ | ✓ | ✓ |

for node-level graph OOD generalization, and test-time training method TENT (Wang et al., 2020); And the recent SOTA *graph data-centric method*: test-time graph transformation method GTRANS (Jin et al., 2023). We use the classic GIN model (Xu et al., 2018) as the well-trained GNN due to its widespread use in molecular graph learning.

### 3.2. Experimental Results

**Test-time GNN Inference Performance.** In Table 1, we compare the ROC-AUC performance of several baseline methods with our proposed PGNDS model on both molecular and protein graph datasets for the graph classification task. In general, we can observe that our proposed PGNDS achieves the highest ROC-AUC across all datasets, indicating the best test-time inference performance for both molecular and protein graphs. Notably, the improvement of PGNDS is significant on the Ogbg-BACE dataset, with a ROC-AUC increase from 0.8338 to 0.8873 over the second-best method. Moreover, TENT provides the second-best results on three datasets except for Ogbg-ClinTox. As TENT is a fine-tuning-based method for adjusting well-trained model parameters, such results demonstrate that the entropy-based fine-tuning can be beneficial for test-time graph classification tasks. Additionally, EERM and GTRANS outperform the other baseline methods on the Ogbg-ClinTox dataset, with EERM securing the second-best result. This can be attributed to its need to adjust the pre-training process to improve GNN generalization ability.

In Table 2, we compare the RMSE performance of various baseline methods with our proposed PGNDS model on molecular graph datasets for the graph regression task. Our PGNDS demonstrates superior performance across all datasets, consistently achieving the lowest RMSE values, which indicates its effectiveness in minimizing regression errors. Specifically, PGNDS shows significant improvements on the QM9-A and QM9-alpha tasks, achieving RMSE reductions from 2.2022 to 2.1985 and from 4.3220 to 4.2881, respectively. This highlights the robustness of PGNDS in handling diverse molecular graph regression tasks. Among the baseline methods, DropEdge shows competitive performance, securing the second-best results on QM9-B, QM9-C, and QM9-alpha tasks, while FeatMask

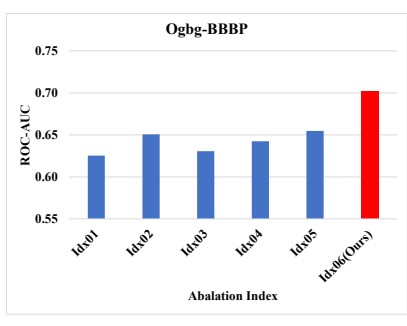

Figure 3. Ablation study results for graph classification (ROC-AUC) on Ogbg-BBBP.

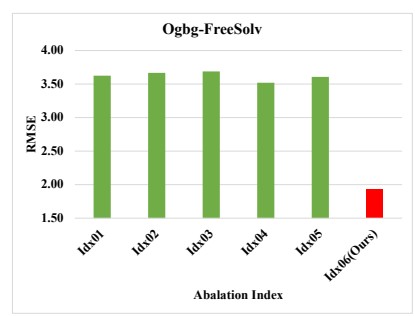

Figure 4. Ablation study results for graph regression (RMSE) on Ogbg-FreeSolv.

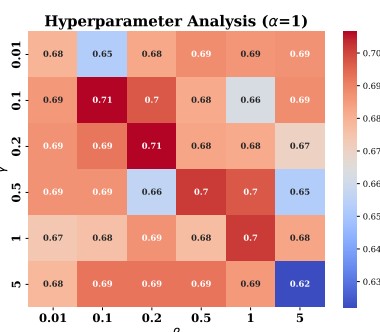

Figure 5. Hyper-parameter sensitivity analysis for $\beta$ and $\gamma$ on Ogbg-BBBP with $\alpha = 1$.

Table 4. Running time (in seconds) comparison on graph classification task in 5 epochs with a single NVIDIA A100 GPU.

| Methods | Ogbg-BBBP | Ogbg-BACE | Ogbg-ClinTox |
|---|---|---|---|
| EERM | 2.784 | 2.287 | 2.224 |
| GTRANS | 0.240 | 0.224 | 0.237 |
| **PGNDS (Ours)** | 2.244 | 2.635 | 2.269 |

performs well on the Ogbg-FreeSolv dataset. We would like to highlight that, compared to the graph classification task, test-time graph regression is often more challenging due to the need for precise fitting of specific target graph properties for regression. This difficulty explains why many test-time methods struggle to deliver satisfactory inference performance, particularly in data-centric approaches that revise the test graphs. Optimizing within such a large latent space to generate specific attributes remains a complex and demanding task.

**Ablation Study of PGNDS.** In Table 3, we list the ablation study settings for evaluating the contributions of different submodules of our PGNDS model. The ablation study systematically removes components from the full model (our PGNDS for Idx06), such as using standard diffusion without guidance (Idx01), the three constraints ($\mathbf{r}_{gtask}$, $\mathbf{r}_{struc}$, and $\mathbf{r}_{gdist}$) in the dual conditional diffusion stage (Idx02 to Idx04), dynamic search (Idx05), and ensemble inference (Idx06). Moreover, it is important to note that these modules interact with one another, and thus, the experimental results do not exhibit linear improvement as the components are added back, highlighting the synergistic effects between the modules. The results are based on ROC-AUC performance for the Ogbg-BBBP classification shown in Fig. 3, and RMSE performance for the Ogbg-FreeSolv regression shown in Fig. 4, respectively. Performance on Ogbg-BBBP demonstrates that as more submodules are activated, the performance steadily improves. The best performance is achieved by Idx06, the full PGNDS model,

with a ROC-AUC of approximately 0.70. In contrast, models with fewer submodules show a lower ROC-AUC. For the Ogbg-FreeSolv dataset, the RMSE values show a similar trend. The model without any guidance has the highest RMSE, indicating poor test-time regression performance. As more components are integrated, the RMSE decreases, with the full model achieving the lowest RMSE.

**Hyper-parameter Sensitivity Analysis.** In Fig. 5, we analyze the hyper-parameter sensitivity for exploring the effects of $\beta$ and $\gamma$ model performance with $\alpha = 1$ for ROC-AUC of Ogbg-BBBP in the dual conditional diffusion procedure for three constraints on $\mathbf{r}_{gtask}$, $\mathbf{r}_{struc}$, and $\mathbf{r}_{gdist}$. The performance is indicated by values across a grid, ranging $\{0.01, 0.1, 0.2, 0.5, 1, 5\}$. More results for $\beta = 1$ and $\gamma = 1$ are presented in Appendix C. The result shows that moderate values of $\beta$, and $\gamma$ generally lead to better test-time inference performance. Extreme values, particularly large $\beta$ or very small $\gamma$, tend to degrade performance slightly.

**Running Time Comparison.** Table 4 compares the running time of different methods on the graph classification task for first-batch test graphs. Our proposed PGNDS demonstrates competitive efficiency, achieving lower runtime than EERM on Ogbg-BBBP and similar performance on Ogbg-ClinTox. While GTRANS remains the fastest due to its constrained learning space with full-parameter transformation of node features and structure, our method balances efficiency and model complexity by integrating test graph distribution remapping, making it an effective graph-level test-time adaptation method. More visualization results of PGNDS generated test-time graph are listed in Appendix D.

## 4. Conclusion

In this work, we address the challenge of test-time adaptation in GNNs by introducing the novel problem of **test-time graph neural dataset search**. Our proposed method, PGNDS, learns a parameterized test-time graph distribution

to improve inference performance. Leveraging dual conditional diffusion, dynamic search, and ensemble inference, PGNDS captures unseen test graph distributions through a generative projection approach. Extensive experiments validate PGNDS superior adaptation capability for graph-level test-time GNN inference. Future research could extend PGNDS to handle dynamic graph neural networks, further broadening its applicability.

## Acknowledgment

Shirui Pan was supported in part by the Australian Research Council (ARC) under grants FT210100097 and DP240101547 and the CSIRO – National Science Foundation (US) AI Research Collaboration Program. Chuan Zhou acknowledged the support from the National Natural Science Foundation of China (No. 62472416). Ming Li acknowledged the support from the National Natural Science Foundation of China (No. 62172370) and the Jinhua Science and Technology Plan (No. 2023-3-003a).

## Impact Statement

This paper presents work whose goal is to advance the field of graph machine learning, specifically in improving the test-time graph inference performance of graph neural networks (GNNs) in a data-centric manner. By proposing a novel test-time graph neural architecture search method, this research seeks to address challenges in deploying GNNs for practical test-time inference. There are many potential societal consequences of our work, none of which we feel must be specifically highlighted here.

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

# Appendix

This is the appendix of our work: **Test-Time Graph Neural Dataset Search With Generative Projection**. Here, we provide additional details on the proposed PGNDS, including more related work discussions, further experimental settings such as dataset statistics, and additional experimental results.

# A. Related Work

**Test-Time Adaptation.** Test-time adaptation (TTA) focuses on dynamically adjusting pre-trained models to improve generalization on test samples (Jin et al., 2023; Wang et al., 2020; Chen et al., 2023a; Liang et al., 2020; Jang et al., 2022), aiming at improving the performance and robustness of deep learning models in the deployment stage (Huang et al., 2021; Huang & Chen, 2022). One of the earliest approaches, test-time training (TTT) (Sun et al., 2020), introduces self-supervised learning with an auxiliary task, enabling model updates using a single test sample. However, TTT requires access to training data to optimize the auxiliary task. To address this limitation, TENT (Wang et al., 2020) proposes a fully test-time adaptation framework, which updates model parameters using only test data, making it more practical for real-world deployment.

*Table A1.* Comparison of different settings for different distribution shift related methods.

| Setting | Source Data | Target Data | Train Loss | Test Loss | Test-time Model Update |
|---|---|---|---|---|---|
| Fine-tuning | - | $\{\mathbf{x}^{(t)}, \mathbf{y}^{(t)}\}$ | $\mathcal{L}(\mathbf{x}^{(t)}, \mathbf{y}^{(t)})$ | - | ✓ |
| Domain Generalization | $\{\mathbf{x}^{(s)}, \mathbf{y}^{(s)}\}$ | any | $\mathcal{L}(\mathbf{x}^{(s)}, \mathbf{y}^{(s)})$ | - | ✗ |
| Domain Adaptation | $\{\mathbf{x}^{(s)}, \mathbf{y}^{(s)}\}$ | $\{\mathbf{x}^{(t)}, \mathbf{y}^{(t)}\}$ | $\mathcal{L}(\mathbf{x}^{(s)}, \mathbf{y}^{(s)}) + \mathcal{L}(\mathbf{x}^{(t)}, \mathbf{x}^{(s)}) + \mathcal{L}(\mathbf{x}^{(t)}, \mathbf{y}^{(t)})$ | - | ✗ |
| Unsupervised Domain Adaptation | $\{\mathbf{x}^{(s)}, \mathbf{y}^{(s)}\}$ | $\mathbf{x}^{(t)}$ | $\mathcal{L}(\mathbf{x}^{(s)}, \mathbf{y}^{(s)}) + \mathcal{L}(\mathbf{x}^{(s)}, \mathbf{x}^{(t)})$ | - | ✗ |
| Test-time Training | $\{\mathbf{x}^{(s)}, \mathbf{y}^{(s)}\}$ | $\mathbf{x}^{(t)}$ | $\mathcal{L}(\mathbf{x}^{(s)}, \mathbf{y}^{(s)}) + \mathcal{L}(\mathbf{x}^{(s)})$ | $\mathcal{L}(\mathbf{x}^{(t)})$ | ✓ |
| Test-time Adaptation (model-centric) | - | $\mathbf{x}^{(t)}$ | - | $\mathcal{L}(\mathbf{x}^{(t)})$ | ✓ |
| **Test-time Adaptation (data-centric)** | - | $\mathbf{x}^{(t)}$ | - | $\mathcal{L}(\mathbf{x}^{(t)})$ | ✗ |

**Test-Time Adaptation on Graph.** Despite the promising performance of TTA, existing methods are predominantly *model-centric* and primarily designed for computer vision (Liang et al., 2024), making them less applicable to GNN models and graph data (Zhang et al., 2024d; Ju et al., 2024; Zhang et al., 2024c; Wu et al., 2024; Chen et al., 2022; Zheng et al., 2024a). Typically, TTA on graphs aims to dynamically fine-tune well-trained GNN models or modify test graphs to improve GNN model generalization and inference performance. Based on the adaptation objective—whether adapting the model or the data—existing methods can be classified into two categories: (a) test-time model adaptation (Chen et al., 2022; Wang et al., 2022; Zhang et al., 2024e;d); and (b) test-time graph adaptation (Jin et al., 2023).

Specifically, given the unseen test graphs, (a) *test-time model adaptation* mainly works on updating the well-trained GNN models using the self-supervised learning paradigm, where the primary objective is to optimize or fine-tune the pre-trained GNN model parameters. For example, GT3 (Wang et al., 2022) and GraphTTA (Chen et al., 2022) employ self-supervised learning techniques to adapt GNN models at test time, targeting graph-level and node-level learning tasks, respectively. Moreover, HomoTTT (Zhang et al., 2024d) proposes a homophily-based and parameter-free graph contrastive learning task for fully test-time GNNs training.

In contrast, (b) *test-time graph adaptation* takes a data-centric approach, modifying the test graph data while keeping the pre-trained GNN model parameters unchanged. Typically, GTRANS (Jin et al., 2023) modifies test graph data without accessing the training procedure or GNN architectures. However, GTRANS relies on a fully parameterized matrix to adjust test-time node features, which restricts its ability to effectively model updated test graphs. Additionally, its binary-space projected gradient descent limits flexibility in handling diverse graph structures. GraphPatcher (Ju et al., 2024) addresses the test-time graph degree shift problem by repatching graphs for node-level tasks. However, its approach is limited to node degree-based modifications, which restricts its flexibility in adapting to diverse test-time distribution shifts at the graph level.

*Table A2.* Dataset statistics for graph classification and regression on protein and molecular graphs.

| Graph Types | Datasets | Task Types | # Graphs | # Tasks | # Nodes Avg./Max | # Edges Avg./Max | # Train / Test |
|---|---|---|---|---|---|---|---|
| Protein | Enzymes | Classification | 587 | 6 | 33.0 / 125 | 63.2 / 149 | 470 / 117 |
| Molecular | Ogbg-BACE | Classification | 1,513 | 1 | 34.1 / 97 | 73.7 / 202 | 1210 / 152 |
| | Ogbg-BBBP | Classification | 2,039 | 1 | 24.1 / 132 | 51.9 / 290 | 1631 / 204 |
| | Ogbg-ClinTox | Classification | 1,477 | 2 | 26.2 / 136 | 55.8 / 286 | 1181/ 148 |
| | Ogbg-FreeSolv | Regression | 642 | 1 | 8.7 / 24 | 16.8 / 50 | 513 / 65 |
| | QM9 | Regression | 133,885 | 4 | 8.8 / 9 | 9.4 /13 | 120803 / 13082 |

In summary, our work addresses the broader challenges of flexibility and generalization in test-time adaptation for GNNs by learning a parameterized test-time graph distribution. This enables improved inference performance on unseen test graphs using well-trained GNNs. Unlike existing graph data-centric TTA methods, our approach captures a wider range of distribution shifts and is particularly effective in handling unknown distribution shifts at the graph level. Furthermore, it is important to highlight why certain baselines were not included in our experiments. Some methods, such as GT3, GraphTTA, and HomoTTT, lack publicly available code, making replication infeasible. Others, like GraphPatcher, operate in fundamentally different learning scenarios that are not directly comparable to our setting. Given these constraints, we carefully selected GTRANS, the most recent state-of-the-art method, as the primary baseline for a fair and meaningful comparison. This ensures that our evaluation remains rigorous and aligned with the intended scope of test-time graph adaptation.

**Distribution Shift Related Methods.** We provide a comparison of different methods addressing distribution shift problems across various learning settings. It outlines the key characteristics for each setting based on the following aspects as shown in Table A1:

- Source Data: The availability of labeled source data during training.

- Target Data: The type of target data used during testing.

- Train Loss: The objective functions used during training, indicating whether source data, target data, or both are used.

- Test Loss: The loss function evaluated during test time, if applicable.

- Test-time Model Update: Whether the model undergoes parameter updates during test time ($\checkmark$ for yes, $\times$ for no).

The table highlights distinctions between:

- Common learning paradigms like fine-tuning, domain generalization, domain adaptation, and unsupervised domain adaptation.

- Test-time training approaches, which incorporate test-time loss evaluation.

- Test-time adaptation, divided into model-centric (updating model parameters) and data-centric (focusing on adapting data rather than modifying model parameters).

The comparison underscores the focus of test-time adaptation methods on improving generalization without relying on access to source data during testing. In this work, we focus primarily on test-time graph adaptation using data-centric methods. Given that different distribution shift related approaches suit varying settings, we excluded comparison methods that pertain to distinct application scenarios.

# B. Dataset Detail

We provide an overview of dataset statistics used for graph classification and regression tasks on protein and molecular graphs in Table A2. The table highlights the diversity of datasets regarding scale, task type, and structural complexity, showcasing their suitability for evaluating test-time graph adaptation methods for graph-level tasks.

## C. Hyper-parameter Analysis

We present more hyper-parameter sensitivity analysis for the Ogbg-BBBP dataset on graph classification, measured using ROC-AUC in Fig. A1. The analysis explores the impact of hyper-parameters $\alpha$, $\beta$, and $\gamma$ in the dual conditional diffusion framework when $\beta = 1$ and $\gamma = 1$.

In general, we can observe that the performance (ROC-AUC) is visualized for varying $\alpha$, $\beta$ $\gamma$ values. And moderate values of $\alpha$, $\beta$, and $\gamma$ are critical for optimal performance, when extreme values tend to degrade the model's effectiveness.

## D. Visualization Comparison

Fig. A2 illustrates a visualization comparison between the original test graph and the proposed PGNDS-generated test graph for molecules in the QM9 dataset for property A. The comparison demonstrates that while key features such as atom connectivity and bond types are preserved, the PGNDS-generated versions exhibit notable differences in graph structure, highlighting the effectiveness of our PGNDS for adapting molecular graphs while maintaining their fundamental chemical properties.

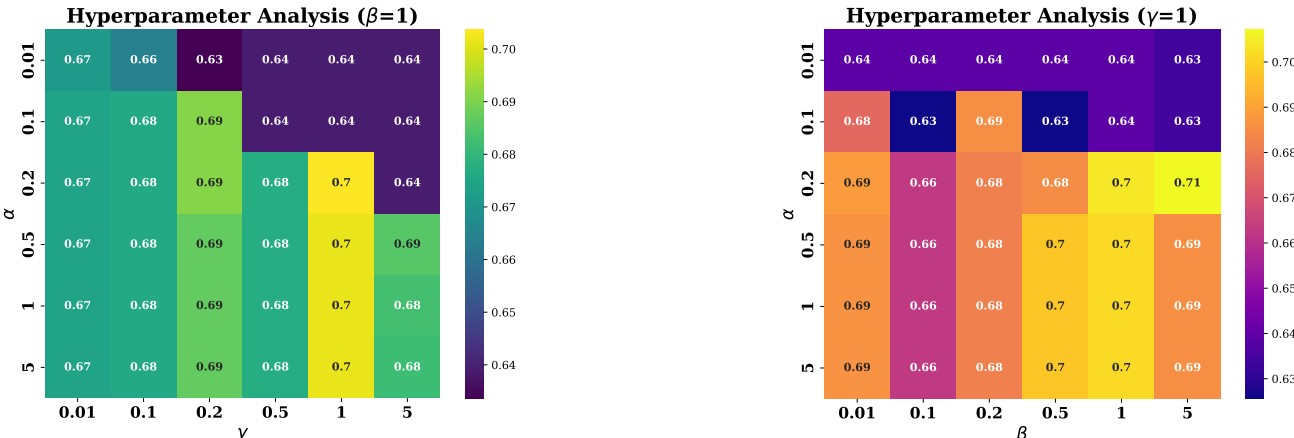

*Figure A1.* Hyper-parameter sensitivity analysis on Ogbg-BBBP for graph classification (ROC-AUC) with $\alpha$, $\beta$, and $\gamma$ in Eq. (12) for different constrains in dual conditional diffusion.

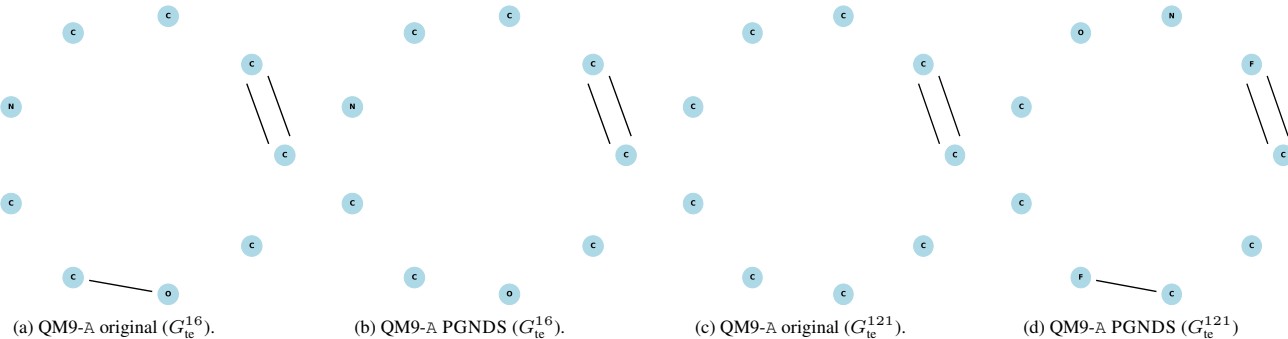

(a) QM9-A original ($G_{\text{te}}^{16}$).  (b) QM9-A PGNDS ($G_{\text{te}}^{16}$).  (c) QM9-A original ($G_{\text{te}}^{121}$).  (d) QM9-A PGNDS ($G_{\text{te}}^{121}$)

*Figure A2.* Visualization comparison between original test graph and our proposed PGNDS generated test graph for molecules in QM9.

