# OpenReview forum: "Test-Time Graph Neural Dataset Search With Generative Projection"
_ICML.cc/2025/Conference — ICML 2025 poster_

### Official Review · Reviewer_AMWj · 2025-03-10

**Overall Recommendation:** 4

**Summary:**

This paper addresses the test-time adaptation challenge in graph neural networks (GNNs).
The main focus is on the generalization about graph data and test-time GNN inference.
The main challenge is that GNN models trained on training graphs may not work well on a new, unseen test graph. The authors propose a new learning problem, test-time graph neural dataset search, along with a generative Projection based test-time Graph Neural Dataset Search method, named PGNDS, to adjust the model at test time so it can better handle new test graphs. PGNDS uses a graph diffusion generator to map new generated test graph to a new distribution close to the training distribution and this process is guided by well-trained GNNs with different constraints for the reverse diffusion process.
The main results of the proposed PGNDS on real-world datasets, show that PGNDS could improve the GNN test time inference results.

**Claims And Evidence:**

Yes, the overall paper is relatively clear, providing detailed descriptions of the proposed method and theoretical justifications.

**Essential References Not Discussed:**

No.

**Experimental Designs Or Analyses:**

Yes, the experimental design is appropriate overall, and the experimental results could verify its effectiveness.

**Methods And Evaluation Criteria:**

Yes, the proposed PGNDS methods and evaluation criteria (six molecular and protein datasets) are reasonable for the problem of test time GNN adaptation and the practical application.

**Other Comments Or Suggestions:**

No.

**Other Strengths And Weaknesses:**

The paper demonstrates several notable strengths.

First, it innovatively propose new test‐time adaptation problem by proposing “graph neural dataset search”. This creative formulation leverages generative projection and dual conditional diffusion to remap unseen test graph distributions toward the training distribution, which is a fresh perspective that could inspire further work in test‐time adaptation for graphs. Second, the modular design (dual conditional diffusion, dynamic search, ensemble inference) is well-structured, and the inclusion of theoretical justifications adds depth. Last, the extensive experimental evaluation—with thorough ablation study and hyperparameter sensitivity analyses—also validate the performance of the proposed PGNDS method on molecular and protein graph tasks.

On the other hand, some potential weaknesses deserve mention.

First, integrating multiple modules (dual conditional diffusion, dynamic search, and ensemble inference) that may impose a high computational overhead and could be challenging to reproduce. Simplifying some aspects or providing additional intuitive explanations might be helpful. Second, the approach assumes access to well-trained GNN and diffusion models, which may not always be feasible.

**Questions For Authors:**

First, given the complexity of the overall framework, will code be release to ensure the reproducibility of the results?

Second, the theoretical justification involving dual conditional diffusion and multiple guidance terms is quite dense. Could you offer a more intuitive explanation of how these components interact in practice to effectively reduce test-time distribution shifts?

Last, how the proposed PGNDS would perform against adversarial perturbations or noisy test-time graphs? Understanding its resilience to noisy data would be valuable for real-world applications.

**Relation To Broader Scientific Literature:**

The key contribution of this paper is to formulate the test-time graph adaptation problem to a new test-time graph dataset generation problem, different from existing test-time adaptation methods on graph, it moves beyond direct model tuning and graph nodes/structure modification.

**Theoretical Claims:**

Yes, the theoretical claims specifically in Section 2.4. Theoretical Justification are checked.

---

> ### Author Rebuttal · Authors · 2025-04-01
>
> **[Re-Weakness(1)] Computation and Explanations of Multiple Module Components in PGNDS**
> >For the three modular components in the proposed PGNDS—dual conditional diffusion, dynamic search, and ensemble inference—we provide a simple one-sentence explanation for each module to help ease understanding:
>
> > - Dual conditional diffusion: a generative process that models test-time distribution shifts via reverse-time diffusion.
> > - Dynamic search: selecting or generating a refined set of test graphs by optimizing over the entire test distribution.
> > - Ensemble inference: aggregating information from the original test graphs and refined adapted graphs to obtain optimal test-time inference performance.
>
> >We have made efforts to clearly structure our paper (e.g., by providing more preliminary background in Section 2.1 and intuitive motivations for each modular design in Section 2.3). We would be happy to clarify any specific concepts that may affect your understanding of our proposed PGNDS.
>
> >Even with these three collaborative modules, we have carefully designed the framework to remain efficient at inference time without introducing substantial overhead. As shown in Table 4, PGNDS achieves comparable or even better runtime than prior baseline methods such as (EERM 2.784s vs. **PGNDS 2.244s** on the OGBG-BBBP dataset). Moreover, our method does not require GNN fine-tuning, and the diffusion steps are truncated ($T \ll T_{\text{tr}}$, refer to Line 128) with early stopping based on the dynamic search criterion in Eq. (12), which significantly reduces unnecessary computation.
>
> ---
>
> **[Re-Weakness(2)] Feasibility of Well-Trained GNN and Diffusion Models**
> >PGNDS is designed to operate under the common and practical test-time adaptation setting, where reasonably well-trained models are available when deployed in practice. This is consistent with prior TTA methods such as TENT and GTRANS, which also rely on pretrained models for test-time refinement. We argue that **this is feasible and reflects a realistic and standard deployment scenario** where well-trained models are commonly used. In this context, PGNDS can serve as a test-time plug-in to further enhance generalization without requiring any model retraining.
>
> ---
>
> **[Re-Questions(1) & (2)] Code and Intuitive Explanation of Dual Conditional Diffusion and Guidance Terms**
> >We will release the code upon the acceptance of this work, considering that it is the first to address the research question of test-time graph neural dataset search. The implementation is not complicated, as it operates entirely at test time without requiring model retraining.
>
> >For an intuitive explanation of dual conditional diffusion and the multiple guidance terms:
> >At a high level, PGNDS uses a reverse-time diffusion process with GNN-provided information as control variables to gradually refine test graphs—i.e., following three conditional guidance terms. Intuitively, they work as follows:
>
> > - First, graph structure preservation (**$r_{\text{struc}}$**) ensures that the refined graph remains close to the original input in structural space, which is important as structural information heavily affects graph properties in test-time graph-level tasks.
> > - Second, task-specific guidance (**$r_{\text{gtask}}$**) encourages the refined graph to align with the pretrained GNN’s predictions (using pseudo labels), thereby improving task relevance.
> > - Third, graph diversity (**$r_{\text{gdist}}$**) prevents the process from collapsing into trivial solutions, maintaining variety in the generated test-time graph candidates.
>
> >During each diffusion step, these constraints jointly influence the latent sampling space, guiding it toward a region in the space that balances test-time graph properties with the joint space defined by [training data \& the well-trained GNNs].
>
> ---
>
> **[Re-Questions (3) ] Robustness to Adversarial Perturbations**
> >Thank you for highlighting this interesting and important research direction. To the best of our knowledge, there is currently **no prior work specifically studying adversarial robustness for test-time graph adaptation methods**, as these methods are inherently unsupervised and operate under the constraint of not modifying deployed models. The field of test-time graph learning is still at an early stage, where the primary focus remains on improving test-time performance under distribution shift.
>
> >Although PGNDS is not explicitly designed for adversarial defense, some components naturally contribute to robustness. For example, during the dynamic search phase, PGNDS generates multiple candidate graphs from the dual conditional diffusion and selects those with better task alignment, which serves as a **filtering mechanism against suboptimal test graphs**. Exploring the robustness of PGNDS under adversarial or noisy settings is indeed a promising direction for future work, and we appreciate the suggestion.

---

### Official Review · Reviewer_TmwA · 2025-03-13

**Overall Recommendation:** 4

**Summary:**

This paper tackles a really interesting and new problem called 'test-time graph neural dataset search.' It enables GNN models handle data they’ve never seen before at test time by creating new graphs similar to the training set. To tackle this problem, the authors propose PGNDS, a method that reconstructs an unseen test graph distribution by leveraging a well-trained GNN as a guiding mechanism. The proposed PGNDS framework contains graph diffusion model as a generator, for generating new test distributions through test-back-to-training distribution mapping, and it uses dynamic search to select new test graphs, and uses new generated graphs and original test graphs to provide final predictions. This work reports the main results and other auxiliary experiments on real-world test graphs.

**Claims And Evidence:**

The authors clearly state their claims, and personally, I find the evidence convincing overall, though some practical implications could be explained more clearly.

**Essential References Not Discussed:**

Not sure, maybe other graph diffusion/generation methods.

**Experimental Designs Or Analyses:**

I briefly checked the soundness/validity of the experimental designs and analyses, as well as the ablation study, hyperparameter analysis, and running time comparison. The experiments look good overall, no issues need to be discussed here.

**Methods And Evaluation Criteria:**

I find the proposed three-stage PGNDS framework interesting and quite practical. Instead of tuning the model itself, it cleverly changes the graph data at test time. The design of PGNDS generally aligns well with the proposed test-time graph neural dataset search problem. The evaluation criteria (ROC-AUC, RMSE) used benchmark datasets (molecular and protein), and baselines, on graph learning tasks are appropriate.

**Other Comments Or Suggestions:**

It would be beneficial to include a more detailed discussion on the limitations of the proposed PGNDS and future potential directions of the proposed test-time graph neural dataset search problem.

**Other Strengths And Weaknesses:**

### Key strengths:
- [Originality] Introduces test-time graph neural dataset search - a novel extension of TTA using generative projection. Shifts focus from graph-level adaptation to distribution-level optimization.
- [Significance] Addresses critical GNN deployment challenge: performance degradation from graph distribution shifts. Practical solution through training distribution projection enhances real-world robustness.-
- [Technical Clarity] Well-structured modular design (dual diffusion, dynamic search, ensemble inference) with rigorous mathematical formulation and theoretical grounding.
- [Validation] Comprehensive experiments across datasets show PGNDS outperforms SOTA baselines. Ablation studies validate component contributions and parameter sensitivity.
### Weaknesses:
- [Complexity] While the proposed generative projection method is innovative, the dual conditional diffusion and dynamic search procedures may introduce significant computational overhead during inference, potentially limiting its applicability in resource-constrained environments.
- [Dependence on Well-trained GNN Models] The effectiveness of PGNDS heavily depends on well-trained pre-existing GNN and diffusion models. This assumes extensive initial training data and reliable model training, whether these assumptions are rational?

**Questions For Authors:**

Here are my questions:

1.The generative projection process seems involves randomness through graph diffusion model, then, under what circumstances might the generative projection process fail or lead to degraded performance?  are there any potential theoretical or practical limitations?

2.The paper introduces ensemble inference as a final step. How is the optimal ratio between original test graphs and newly generated graphs determined for achieving the best inference results?

3.The paper claims that the test-time graph distribution is mapped back to the training distribution using a diffusion model. However, is there a formal guarantee that the generated graphs truly align with the training distribution in a way that improves generalization, rather than simply generating more training-like graphs that may not be representative of the test distribution?

**Relation To Broader Scientific Literature:**

This work broadly discusses its connection to the graph distribution shift problem in the supplementary material, particularly in “Table A1, Comparison of different settings for different distribution shift related methods”. The relationship between this work and other research methods is relatively clear and helps distinguish it from existing approaches.

**Theoretical Claims:**

I quickly checked the theoretical parts, and they seem logical. However, it would help if the authors could simplify or clarify the intuition behind these theories.

---

> ### Author Rebuttal · Authors · 2025-04-01
>
> **[Re-Weakness(1)] Computation of Dual Conditional Diffusion and Dynamic Search**
> >While PGNDS introduces dual conditional diffusion and dynamic search, we have carefully designed the framework to remain efficient at inference time **without introducing substantial overhead**. As shown in Table 4, PGNDS achieves comparable or even better runtime than prior baseline methods such as EERM:2.784s vs Ours PGNDS: 2.244s on Ogbg-BBBP dataset. Moreover, our method does not require well-trained GNN fine-tuning, and the diffusion steps are truncated ($T \ll T_{tr}$ refer to Line 128) with early stopping based on the dynamic search criterion in Eq. (12), which significantly reduces unnecessary computation.
>
> ---
> **[Re-Weakness(2)] Well-trained GNNs and Diffusion Model**
> >PGNDS is designed to operate under the common and practical test-time adaptation setting, where reasonably well-trained models are available when deployed in practice. This is consistent with prior TTA methods such as TENT and GTRANS, which also rely on pretrained models for test-time refinement. We argue that this is **not an assumption, but rather a realistic and standard deployment scenario** where well-trained models  are commonly used. In this context, PGNDS can serves as a test-time plug-in to further enhance generalization without requiring any model retraining during test time.
>
> ---
>
> **[Re-Other Comments Or Suggestions] Limitations and Future Directions**
> >Thank you for the constructive suggestion. One potential limitation of PGNDS is that it currently focuses on graph-level tasks and relies on task-specific guidance derived from model predictions. For instance, extending the current design to node-level settings requires adapting the task-specific guidance term $r_\text{gtask}$ with careful design, especially to ensure compatibility with the structure and graph diversity constraints in Eq. (10).
>
> >Looking forward, the proposed test-time graph neural dataset search paradigm opens up several promising directions, such as (1) adapting to **more diverse graph types and GNN types (e.g., dynamic graphs and dynamic GNNs)** and (2) integrating PGNDS with **continual or online learning frameworks under open-world scenarios**. We will include a more detailed discussion on these aspects in the final version.
>
> ---
>
> **[Re-Questions(1)] Under What Circumstances the Generative Projection Process Might Fail**
>
> >While PGNDS leverages a stochastic generative diffusion process to refine test-time graphs, this randomness is **an advantage of our proposed PGNDS, as it provides a broader search space for learning meaningful test-time graph candidates** and does not degrade model performance in practice. One potential limitation that might affect the generative projection process arises under severe distribution shifts, where the test-time graph distribution is highly dissimilar from the training distribution. In such cases, the reverse-time projection may produce low-quality or less informative graphs due to insufficient alignment. Nonetheless, our empirical results (Table 1 and Table 2) demonstrate that PGNDS remains robust across a wide range of realistic test-time scenarios.
>
> ---
>
> **[Re-Questions(2)] Ensemble Inference Ratio**
> >In our current implementation, we adopt a 1:1 ratio between the original test graphs and the adapted graphs during ensemble inference, as defined in Eq. (13). This fixed ratio is both simple and effective, and consistently yields improved performance across datasets, as shown in Table 2 and Table 3. We are open to solutions like learning or tuning adaptive weights; it is not hard to implement but would introduce extra learning parameters and further complicate the method.
>
> ---
>
> **[Re-Questions(3)] Distribution Alignment and Generalization Guarantee**
> >While PGNDS uses a diffusion model to project the test-time graph distribution toward the training distribution, **the goal is not to perfectly change the test graphs to the training graphs, but rather to map test graphs into a joint space defined by [the training distribution \& the pretrained GNNs].** This process preserves the intrinsic properties of test graphs while enhancing their compatibility with the well-trained GNN’s learned decision boundary on the training graphs.
>
> >In other words, PGNDS performs a model-aware distribution alignment, guided by three constraints in Eq. (10), to improve generalization. This is also supported theoretically in Proposition 2.3 with Eq. (18) - (20), where we show a bounded deviation in the refined distribution, and empirically in Table 2 and Table 3, where the refined graphs consistently yield better predictions.

---

### Official Review · Reviewer_VWGF · 2025-03-13

**Overall Recommendation:** 3

**Summary:**

This paper introduces test-time graph neural dataset search with generative projection to improve test-time adaptation for Graph Neural Networks (GNNs) facing distribution shifts. The proposed method, PGNDS, uses a generative projection approach to refine test graphs without modifying the trained GNN. PGNDS consists of three key steps: dual conditional diffusion for graph generation, dynamic search for selecting the best test graphs, and ensemble inference for improved predictions. Experiments on real-world graph datasets shows better performance over baselines.

**Claims And Evidence:**

Yes,  no problematic claims are found.

**Essential References Not Discussed:**

N/A.

**Experimental Designs Or Analyses:**

Generally checked.

**Methods And Evaluation Criteria:**

Yes, make sense.

**Other Comments Or Suggestions:**

See the previous question.

**Other Strengths And Weaknesses:**

For strengths:

(1)The paper introduces a new learning problem for handling test-time adaptation in graph data, which appears to be a novel challenge for GNN generalization at inference time. The idea of adapting test graphs without modifying model parameters seems practical for real-world applications.

(2)The overall writing is organized well, the background about test-time adaptation on graphs and challenges is described relatively clearly. Each module of PGNDS method, i.e., leveraging dual conditional diffusion, dynamic search, and ensemble inference, are well-designed and well-described, although it is a little complex to understand.

(3)This paper includes several experiments that show the proposed method performs better than existing approaches, showing the proposed PGNDS could capture unseen test graph distributions through a generative method guided by well-trained GNNs.

For weaknesses:

(1)This method involves many concepts like graph diffusion, conditional guidance, and dataset search, it is quite difficult to fully understand it.

(2)How does this work relate to graph neural architecture search? should this be discussed?

(3)Some parts, like how the "dynamic search" process selects useful graphs, are not fully explained, more details need to be explained further. How to stop the search and guarantee the selected test graphs are optimal?

**Questions For Authors:**

See the weakness.

Another question is about the generalization ability of the model. The evaluation primarily focuses on molecular graphs for graph classification and regression tasks. Would the proposed PGNDS be equally effective on other types of graphs, such as social networks or citation graphs, or on different graph learning tasks, like node classification? If so, could you explain how PGNDS would adapt to these different graph types and tasks?

**Relation To Broader Scientific Literature:**

Relatively related to existing graph learning problem and GNNs.

**Theoretical Claims:**

Did not go into much detail, but generally checked.

---

> ### Author Rebuttal · Authors · 2025-04-01
>
> We thank the reviewers for highlighting **the novelty and practicality of our proposed test-time graph neural dataset search (PGNDS)**, as well as the **clear organization and strong empirical results**. Detailed responses regarding the more key concept explaination, the dynamic search process and stopping criterion, and the method’s generalizability across diverse tasks and datasets are provided below.
>
> ---
>
> **[Re-Weakness-(1)] On Method Built on Multiple Concepts**
> >Thank you for the feedback. We understand that the proposed PGNDS introduces multiple concepts—such as graph diffusion, conditional guidance, and dataset-level search—which may be challenging to follow at first. We provide a simple one-sentence explanation for each concept to help ease understanding:
>
> > - Graph diffusion: a generative process that models test-time distribution shifts via reverse-time diffusion;
> > - Conditional guidance: a mechanism to control the reverse-time diffusion for generating refined test graphs using pretrained GNNs' knowledge;
> > - Dataset-level search: selecting or generating a refined set of test graphs by optimizing over the entire test distribution.
>
> >We have made efforts to clearly structure our paper (e.g., provide more preliminary background in Section 2.1 and intuitive motivations for each modular design in Section 2.3). We would be happy to clarify any specific concepts that may affect your understanding of our proposed PGNDS.
>
> ---
>
> **[Re-Weakness-(2)] On Relation to Graph Neural Architecture Search (GNAS)**
> >Thank you for the question. Our test-time graph dataset search (GNDS) and GNAS have fundamentally different objectives. GNAS aims to **search for optimal GNN architectures** (e.g., layers, aggregators), typically during the training stage. In contrast, our method PGNDS focuses on test-time dataset-level graph refinement—**searching for optimal test graphs** to improve prediction under distribution shift, without modifying the model architecture. Therefore, our work serves a completely different purpose from GNAS.
>
> ---
>
> **[Re-Weakness-(3)] On Dynamic Search and Stopping Criterion**
> >The dynamic search process in PGNDS iteratively samples refined graphs from the conditional diffusion-based candidate set in Eq. (11) and evaluates them using a test-time rectification objective $\epsilon$ in Eq. (12). For each test graph, we select the adapted version that minimizes this objective across the reverse diffusion steps. To ensure search efficiency and stability, we adopt a **dynamic stopping strategy**: the process halts early if the objective does not improve over a certain number of patience steps (refer to Lines 267–270 in Section 2.4), thus avoiding exhaustive sampling. The effectiveness of this guided selection is also supported by our empirical study—Table 3 (Idx04 vs. Idx05) in Fig.3 and Fig.4 shows that it consistently identifies high-quality test graphs.
>
> ---
> **[Re-Questions For Authors] On Generalization to Other Graph Types and Tasks**
> >Thank you for the thoughtful question. In this work, we focus on molecular and protein graphs with graph-level tasks, **covering six benchmark datasets and two types of graph learning tasks (classification and regression, with nine detailed tasks)**. This demonstrates that PGNDS is effective across a wide range of real-world graph-level scenarios. We would like to emphasize that **the core framework of PGNDS is task-agnostic and generalizable**. For example, adapting PGNDS to node classification tasks (e.g., on other graph types, i.e., citation or social networks) would involve redefining the task-specific guidance term $r_{\text{gtask}}$ to reflect node-level objectives, and the conditional diffusion and structural constraints remain applicable. We believe the framework’s modular design makes such extensions straightforward.

---

> > ### Comment · Reviewer_VWGF · 2025-04-06
> >
> > Thank you for the author's response. I have also reviewed the comments from the other reviewers and the corresponding replies from the author. I will maintain my score.

---

### Official Review · Reviewer_G3YQ · 2025-03-14

**Overall Recommendation:** 3

**Summary:**

The authors introduce a new problem, test-time graph neural dataset search, to learn the optimal distribution of unknown test graph datasets. For this purpose, they propose PGNDS, a generative projection driven by a diffusion model. By projecting test graphs back to the training distribution, PGNDS learns test-time adaptation by generating refined test graphs. Experimental results demonstrate the effectiveness of PGNDS.

**Claims And Evidence:**

What is the specific difference between the problem “test-time graph neural dataset search” and “test-time graph adaptation”?


Would projecting the entire test graph set distribution back to the training set change the properties of refined test graphs?

**Essential References Not Discussed:**

None

**Experimental Designs Or Analyses:**

The “ensemble inference” phase aggregates the representative information from both the original and adapted test graphs. Experiments should their individual results, and explain why the method does not use only the adapted graph. Because the adapted graph seems to be the best structure.

**Methods And Evaluation Criteria:**

This method is applicable to small molecular and protein graphs. Is it applicable to larger graphs?

**Other Comments Or Suggestions:**

This paper should compare more recent methods, including test-time model adaptation methods and graph adaptation methods they refer to.

**Other Strengths And Weaknesses:**

Strengths: The proposed method is novel for test-time graph-level tasks.

Weaknesses: Recent methods are not compared in the experiment.

**Questions For Authors:**

In figure A2, I cannot understand the effect of graph structure adaption. Why (b) and (d) is better than (a) and (c)?

**Relation To Broader Scientific Literature:**

Compared with the test-time model adaptation methods, this paper avoids modifying GNN parameters and is more applicable. This paper also proposes an innovative method.

**Theoretical Claims:**

The theoretical analyses seem to be correct.

---

> ### Author Rebuttal · Authors · 2025-04-01
>
> **[Re-Claims and Evidence (1): Difference between “Test-Time Graph Neural Dataset Search” and “Test-Time Graph Adaptation”]**
>
> >We thank the reviewer for recognizing our contribution in proposing the **novel problem of test-time graph neural dataset search (test-time GNDS)**. In brief, **test-time GNDS can be viewed as a distribution-level extension of traditional test-time graph adaptation**, where instead of modifying each test graph individually, we **learn a parameterized graph distribution at the dataset level**. From a broader perspective, both approaches are data-centric test-time graph manipulation methods. We summarize the key differences in the following table, and we will further clarify this conceptual distinction in the revised version.
>
> >| Aspect                         | Test-Time Graph Adaptation                         | Test-Time Graph Neural Dataset Search      |
> >|-------------------------------|----------------------------------------------------|------------------------------------------------------|
> >|Granularity               | Per-graph (adapts each test graph individually)    | Dataset-level (searches a distribution over graphs)  |
> >|Adaptation Objective      | Modify a single test graph to improve prediction locally  | Generate task-aligned graph distributions toward training domain  |
> >| Method Type               | Direct feature and structure manipulation | Generative modeling + sampling via diffusion         |
> >| Optimization Target       | One graph at a time                                | Entire test-time graph set as a distribution         |
>
> ---
> **[Re-Claims And Evidence (2)] Whether test-time distribution projection change the properties of refined test graphs?**
>
> > PGNDS projects the test-time graph distribution toward the training, but it is **not to overly change** the intrinsic properties of test graphs. This is achieved via the **dual conditional diffusion process** with three key constraints in Eq. (10): (1) **r_struc**: structure preservation; (2) **r_gdist**: graph diversity control; and (3) **r_gtask**: task-specific preservation. These constraints ensure that the refined test graphs remain closely related to the original inputs while gaining improved compatibility with the training distribution.
>
> >Besides, as shown in Proposition 2.3 and Eq. (20), the deviation from the original distribution is bounded by a controllable term $|\xi|$, guided by GNN-specific knowledge. Moreover, empirical results (Table 3, Fig. 3–4) confirm that removing these constraints significantly degrades performance, validating the role of dual conditional diffusion in preserving test graph properties while improving inference.
>
> ---
>
> **[Re-Methods and Evaluation Criteria] Applicability to Larger Graphs**
> >While our experiments focus on small molecular and protein graphs, PGNDS is not inherently limited to such settings. We evaluate PGNDS on the large-scale QM9 dataset, which contains 133,885 molecular graphs, demonstrating its applicability to large test sets. As a graph-level method, PGNDS can be applied to graphs with more nodes as well, with scalability primarily depending on the efficiency of the diffusion model.
>
> ---
> **[Re-Experimental Designs or Analyses] Ensemble Inference**
> >As confirmed in our ablation study (Table 3, Idx05: only adapted graphs vs. Idx06: ensemble of original and adapted graphs), ensemble inference consistently outperforms using adapted graphs alone. While the adapted test set is aligned with the training distribution, the test-time distribution learning remains inherently complex. Due to the large search space and potential approximation errors in the generation process [refer to Lines 265-272], relying solely on adapted graphs may not capture all informative patterns. Therefore, we propose an ensemble inference scheme to aggregate complementary information from both original and adapted graphs.
>
> ---
>
> **[Re-Weakness and Suggestions] Recent methods.**
> >Our work is **the first to formulate the problem of test-time graph-level dataset search** with generative projection. To the best of our knowledge, no existing method directly addresses this setting. While we include representative baselines from both test-time model adaptation (e.g., TENT) and graph adaptation (e.g., GTRANS), there are currently no more recent or directly comparable methods for our proposed task. We will continue to monitor future developments and further explore this new research direction.
>
> ---
>
> **[Re-Questions For Authors] On Figure A2**
>
> >Compared to the originals in (a) and (c), the adapted graphs in (b) and (d) exhibit more chemically meaningful substructures (e.g., double bonds).**This does not imply that the adapted graph is a “better” version in a ground-truth sense**, but rather that it is **the most aligned with the training distribution under a well-trained model**, and such alignment is beneficial for improving test-time inference performance.

---

### Decision · Program_Chairs · 2025-05-01

**Decision:**

Accept (poster)

**Comment:**

This paper introduces a novel and timely framework, PGNDS, which tackles the problem of test-time adaptation for Graph Neural Networks (GNNs) through a new paradigm termed test-time graph neural dataset search. Instead of adapting models or individual graphs, PGNDS performs distribution-level test graph refinement using a generative diffusion process guided by pretrained GNNs. The framework is modular and theoretically grounded, incorporating dual conditional diffusion, dynamic search, and ensemble inference. Reviewers largely agree on the significance and originality of the problem formulation, the technical soundness of the proposed solution, and the comprehensiveness of the experimental validation. While some reviewers noted the complexity of the framework and the need for clearer explanations or expanded evaluation, the authors provided detailed and thoughtful rebuttals that addressed these points effectively. Given its originality, well-executed design, and strong empirical performance, I recommend acceptance.